# Testing Interdependent Self-Construal as a Moderator between Mindfulness, Emotion Regulation, and Psychological Health among Emerging Adults

**DOI:** 10.3390/ijerph18020444

**Published:** 2021-01-08

**Authors:** Ming Chen, Rebecca Y. M. Cheung

**Affiliations:** 1Department of Mathematics and Information Technology, The Education University of Hong Kong, Hong Kong, China; s1112014@s.eduhk.hk; 2Department of Early Childhood Education, Centre for Child and Family Science, and Centre for Psychosocial Health, The Education University of Hong Kong, Hong Kong, China

**Keywords:** depressive symptoms, emotion regulation, interdependent self-construal, life satisfaction, mindfulness

## Abstract

This study examines the moderating role of interdependent self-construal between mindfulness, emotion regulation, and psychological health, with emotion regulation as a mediator. A total of 187 Chinese emerging adults completed self-reported measures, including mindfulness, cognitive reappraisal, expressive suppression, depressive symptoms, life satisfaction, and interdependent self-construal. Our findings indicate moderation effects of interdependent self-construal between (i) mindfulness and cognitive reappraisal, (ii) cognitive reappraisal and life satisfaction, (iii) expressive suppression and life satisfaction, and (iv) expressive suppression and depressive symptoms. Based on bootstrapping and path analyses, cognitive reappraisal mediated the relation between mindfulness and psychological health, including depressive symptoms and life satisfaction, regardless of the level of interdependent self-construal. In addition, mindfulness was not related to expressive suppression, regardless of the level of interdependent self-construal. Based on these findings, researchers and practitioners should pay attention to the differential associations between mindfulness, emotion regulation strategies, and psychological outcomes as a function of interdependent self-construal during emerging adulthood.

## 1. Introduction

Mindfulness refers to individuals’ capacity to observe and pay full attention to their ongoing experiences without judgment [1]. Previous studies have shown that mindfulness is linked to better psychological health, including greater life satisfaction, lower distress and perceived stress, and lower social anxiety [2,3,4]. Mindful individuals are prone to accepting their experiences, thoughts, and emotions non-reactively and non-judgmentally. To understand why and how mindfulness is related to psychological health, researchers argue that emotion regulation is a crucial process [3,5,6]. As correlates of mindfulness, people’s non-judgmental awareness and attentiveness of their experiences facilitate adaptive emotion regulation [7]. By having lower negative emotional reactivity and better emotion regulation skills, they are more likely to have better mental health and quality of life [3,8,9].

Cognitive reappraisal and expressive suppression have been commonly studied as two major forms of emotion regulation strategies [10,11]. Cognitive reappraisal takes the form of rethinking the meaning of stimuli (e.g., an event) or distancing oneself from thoughts to perceive facts from a third-person perspective [10]. Through cognitive reappraisal, people sometimes generate new insight to reinterpret events in a more positive way. For example, in the face of an event initially viewed as a threat, an individual may objectively reinterpret the situation as an opportunity for personal development, regardless of his or her feelings and thoughts resulted from the initial appraisal. Expressive suppression, on the other hand, involves withholding or modifying behavioral expressions of emotions without changing the thoughts, felt emotions, or emotional experiences [10]. As common emotion regulation strategies, expressive suppression and cognitive reappraisal differ on the extent to which people engage in cognitive change versus inhibition of emotional expressions.

Previous studies have shown that mindfulness is positively linked to cognitive reappraisal [12]. According to the mindful coping model [13], mindful individuals are likely to reappraise situations positively as a result of greater attentional broadening. Specifically, mindfulness enhances people’s capability to decenter from negative emotions and distressing thoughts, such that they can evaluate events from multiple perspectives. According to research evidence on brain activity [12,14], mindfulness is positively associated with cognitive reappraisal. Specifically, mindfulness and cognitive reappraisal share some neural mechanisms, such as the regulating structures and the target regions of amygdala [14]. Studies conducted in various cultural contexts also suggested that people with greater mindfulness are more likely to reappraise events in a positive light [15].

Mindfulness facilitates expressive behaviors that are consistent with individual values [16]. As such, mindfulness may lessen expressive suppression, as mindful individuals are more likely to accept emotional experiences without judgment [17]. Through non-judgmental awareness and acceptance, people may be more comfortable with expressing their emotions. Supporting the association between mindfulness and expressive suppression, previous research suggested that participants of mindfulness-based stress reduction programs had a significant decrease in expressive suppression [18]. Likewise, Asian American and Latino American adolescents participating in a schoolwide mindfulness-based intervention also reported a greater reduction of expressive suppression compared to their control group counterparts [19]. Altogether, recent findings have evidenced the association between mindfulness and emotion regulation strategies, including cognitive reappraisal and expressive suppression.

The association between emotion regulation and psychological well-being has long been established. Of note, cognitive reappraisal is associated with better mental health and fewer distress symptoms [11]. Cognitive reappraisal also fosters reductions in negative emotional experience and behavioral expression without engendering much physiological cost [20]. It is also linked to greater levels of savoring positive experiences, fewer negative emotions, and fewer symptoms of depression and anxiety [10,15,21]. In contrast, expressive suppression incurs substantial physiological and psychological costs. Notably, the incongruence between inner arousal and outer expression may lead to greater mental energy consumption [22] and more negative emotional experiences about oneself and others [10]. To make matters worse, suppressors are less likely to receive social support, given their hidden emotional experiences and needs. Consequently, expressive suppression is linked to poorer life satisfaction, greater psychological distress, and more depressive symptoms [21].

Returning to the mindful coping model [13], positive reappraisal is a process between mindfulness and reduced stress. Supporting its theoretical tenets, empirical findings showed that cognitive reappraisal mediated between mindfulness and psychological distress [23,24]. Expressive suppression also mediated between mindfulness and depression [24]. As such, both cognitive reappraisal and expressive suppression are important processes through which mindfulness fosters psychological health.

### 1.1. Does Interdependent Self-Construal Play a Role in These Processes?

Emerging evidence shows that factors such as cultural value or culture of origin can moderate the relation between emotion regulation and psychological functioning [25,26]. According to the conceptual framework of self-construal [27], people with high interdependent self-construal are more concerned about interpersonal harmony over their own emotional needs, such that other people’s thoughts and emotions are prioritized. As such, emotion regulation strategies such as expressive suppression may be less detrimental for those who view themselves as highly interdependent, given that they are oriented towards others’ needs and concerns [25,28]. For instance, previous research showed that interdependent self-construal moderated between anger suppression and depressive symptoms, such that the association between anger suppression and depression was attenuated for people with greater interdependent self-construal [25]. Contrary to expressive suppression, however, little is known about the moderating role of interdependent self-construal on the path between cognitive reappraisal and psychological adjustment, although the mean levels of cognitive reappraisal appear to be similar across cultures [29]. Likewise, little has been done to examine the moderating role of interdependent self-construal between mindfulness and emotion regulation strategies.

### 1.2. The Present Study

In this study, we hypothesized that mindfulness would be associated with emotion regulation strategies. As mediators, emotion regulation strategies would then be associated with life satisfaction and depressive symptoms. We further hypothesized that interdependent self-construal would moderate the hypothesized associations. For individuals with greater interdependent self-construal, the relation between suppression and psychological health (i.e., life satisfaction and depressive symptoms) would be attenuated. For the other paths, we did not specify an a priori hypothesis, given the lack of previous findings concerned with the moderating effect of interdependent self-construal. Figure 1 depicts the hypothesized model.

## 2. Materials and Methods

### 2.1. Participants

A total of 187 emerging adults were recruited from a public university in Hong Kong, ranging from 18 to 25 years of age (M = 21.60; SD = 1.88; 89.84% female, *n_female_* = 168). Participants were recruited from mass emails and online platforms. The study was approved by the Human Research Ethics Committee of the authors’ institution. Prior to the commencement of the study, informed consent was sought, and participants were told that they could withdraw from the study at any time or decide not to respond to any of the questions. Participants received a $50 coupon from a local supermarket (~US $6.41) to compensate for their time.

### 2.2. Measures

The measures were translated from English to Chinese using back-translation procedures [30]. The research team discussed and resolved the discrepancies arising from the translation.

#### 2.2.1. Mindfulness

The 10-item cognitive and affective mindfulness scale-revised (CAMS-R) [1] was used to measure everyday mindfulness and the extent to which participants experienced their thoughts and feelings. The subscales included present focus, attention, acceptance, as well as awareness. The items were rated on a 4-point scale from 1 (rarely or not at all) to 4 (almost always). Higher averaged scores indicate greater mindfulness. In this study, Cronbach’s alpha was equal to 0.82.

#### 2.2.2. Emotion Regulation

The 10-item emotion regulation questionnaire (ERQ) [10] was used to measure emotion regulation strategies, including cognitive reappraisal and expressive suppression, on a 7-point scale from 1 (strongly disagree) to 7 (strongly agree). Higher averaged scores indicated more cognitive reappraisal and expressive suppression, respectively. In this study, Cronbach’s alpha was equal to 0.83 for cognitive reappraisal and 0.73 for expressive suppression. Although Cronbach’s alpha was less than 0.80 for expressive suppression, an alpha larger than 0.70 is regarded as acceptable in psychological constructs [31].

#### 2.2.3. Life Satisfaction

The 5-item satisfaction with life scale (SWLS) [32] was used to measure life satisfaction on a 7-point scale from 1 (totally disagree) to 7 (totally agree). Higher averaged scores indicated greater life satisfaction. In this study, Cronbach’s alpha was equal to 0.87.

#### 2.2.4. Depressive Symptoms

The 9-item patient health questionnaire depression module (PHQ-9) [33] was used to measure depressive symptoms on a 4-point scale ranging from 0 (not at all) to 3 (almost every day). The items assessed depressive thoughts, fatigue, and concentration in the previous two weeks. Higher averaged scores indicated greater severity of depression. In this study, Cronbach’s alpha was equal to 0.87.

#### 2.2.5. Interdependent Self-Construal

The 14-item interdependent self-construal subscale of the self-construal scale (SCS) [34] was used. Participants rated on a 5-point scale ranging from 1 (strongly disagree) to 5 (strongly agree). Higher averaged scores indicated greater interdependent self-construal. In this study, Cronbach’s alpha was equal to 0.85.

### 2.3. Data Analysis

Path analysis was conducted to examine the fit of the hypothesized model to the data. Full information maximum-likelihood estimation was used to treat the missing data. To facilitate interpretability, mindfulness, expressive suppression, cognitive reappraisal, and interdependent self-construal were centered to the mean. To examine the moderating effect of interdependent self-construal between mindfulness and emotion regulation strategies, interdependent self-construal, mindfulness, and the interaction term between these predictors were entered into the first path model. Bootstrapping was then used to investigate the mediation effects of emotion regulation strategies. Next, to examine the moderating effect of interdependent self-construal between emotion regulation strategies and psychological outcomes, a second path model was conducted. Specifically, interdependent self-construal, expressive suppression, cognitive reappraisal, and the two interaction terms between interdependent self-construal and the emotion regulation strategies were entered. Bootstrapping was then used to investigate the mediation effects of emotion regulation strategies and the interaction terms between mindfulness and psychological outcomes. When the moderation effects were significant, post hoc simple slopes test [35] was used to further evaluate the differences between the slope coefficients when interdependent self-construal was low (i.e., 1 SD below the mean) vs. high (i.e., 1 SD above the mean).

## 3. Results

Table 1 shows the correlations, means, and SDs for all variables.

### 3.1. Testing Interdependent Self-Construal as a Moderator between Mindfulness and Emotion Regulation

In the first set of path analysis, a test of emotion regulation strategies as mediators between exogenous predictors (i.e., interdependent self-construal, mindfulness, and their interaction term) and psychological outcomes (i.e., life satisfaction and depressive symptoms) yielded good fit to the data, χ^2^(3) = 6.66, *p* > 0.05, CFI = 0.98, RMSEA = 0.08, SRMR = 0.04. Specifically, mindfulness (β = 0.25, *p* < 0.001), interdependent self-construal (β = 0.31, *p* < 0.001), and the interaction term between mindfulness and interdependent self-construal (β = −0.15, *p* = 0.02) predicted cognitive reappraisal. Post hoc simple slopes test showed that the slope was significantly different from zero when interdependent self-construal was low, but not when it was high. That is, the relation between mindfulness and cognitive reappraisal revealed a significantly positive relation when interdependent self-construal was low (i.e., 1 SD below the mean), β = 0.39, *p* < 0.001, but not when it was high (i.e., 1 SD above the mean), *p* > 0.05 (see Figure 2).

Expressive suppression was not related to mindfulness, interdependent self-construal, or the interaction term, *p*s > 0.05. Life satisfaction was predicted by cognitive reappraisal (β = 0.29, *p* < 0.001), mindfulness (β = 0.14, *p* = 0.04), and interdependent self-construal (β = 0.25, *p* < 0.001), but not by expressive suppression or the interaction between mindfulness and interdependent self-construal, *p*s > 0.05. Depressive symptoms were predicted by cognitive reappraisal (β = −0.17, *p* = 0.009), expressive suppression (β = 0.28, *p* < 0.001), and mindfulness (β = −0.37, *p* < 0.001), but not by interdependent self-construal or the interaction between mindfulness and interdependent self-construal, *p*s > 0.05.

Drawing from the above findings, cognitive reappraisal was examined as a mediator via bootstrapping. Based on 5000 bootstrap samples with replacement, the 95% CI indicated that the standardized indirect effects of the interaction between mindfulness and interdependent self-construal on life satisfaction (CI: (−0.10, 0.000)) and depressive symptoms (CI: (−0.001, 0.08)) via cognitive reappraisal included zeros. Therefore, although mindfulness interacted with interdependent self-construal to predict cognitive reappraisal, the moderated mediation effect was not supported. However, based on 5000 bootstrap samples with replacement, the 95% CI indicated that the standardized indirect effects of mindfulness on life satisfaction (CI: (0.03, 0.14)) and depressive symptoms (CI: (−0.11, −0.004)) via cognitive reappraisal did not include zeros, thereby supporting cognitive reappraisal as a mediator.

### 3.2. Testing Interdependent Self-Construal as a Moderator between Emotion Regulation and Psychological Health

The second path model involving emotion regulation strategies, interdependent self-construal, and the two interaction terms between emotion regulation strategies and interdependent self-construal yielded good fit to the data, χ^2^(9) = 13.37, *p* > 0.05, CFI = 0.98, RMSEA = 0.05, SRMR = 0.04. Specifically, mindfulness predicted cognitive reappraisal (β = 0.32, *p* < 0.001), but not expressive suppression or the two interaction terms between interdependent self-construal and emotion regulation strategies, *p*s > 0.05. Life satisfaction was predicted by mindfulness (β = 0.14, *p* = 0.03), cognitive reappraisal (β = 0.27, *p* < 0.001), interdependent self-construal (β = 0.24, *p* < 0.001), the interaction between interdependent self-construal and cognitive reappraisal (β = −0.14, *p* = 0.03), and the interaction between interdependent self-construal and expressive suppression (β = −0.14, *p* = 0.02), but not expressive suppression alone, *p* > 0.05. According to post hoc simple slopes test, the relation between cognitive reappraisal and life satisfaction was more significantly positive when interdependent self-construal was low (i.e., 1 SD below the mean), β = 0.43, *p* < 0.001, but less significantly positive when it was high (i.e., 1 SD above the mean), β = 0.22, *p* = 0.02 (see Figure 3).

As for the link between expressive suppression and life satisfaction, the post hoc simple slopes test showed that the relation between expressive suppression and life satisfaction was significantly negative when interdependent self-construal was high (i.e., 1 SD above the mean), β = −0.27, *p* = 0.003, but not significant when it was low (i.e., 1 SD below the mean), *p* > 0.05 (see Figure 4).

Finally, depressive symptoms was predicted by mindfulness (β = −0.35, *p* < 0.001), cognitive reappraisal (β = −0.20, *p* < 0.01), expressive suppression (β = 0.29, *p* < 0.001), and the interaction between interdependent self-construal and expressive suppression (β = −0.19, *p* = 0.01), but not by interdependent self-construal alone or the interaction between interdependent self-construal and cognitive reappraisal, *p*s > 0.05. Post hoc simple slopes test showed that the slope was significantly different from zero when interdependent self-construal was low, but not when it was high. Specifically, the relation between expressive suppression and depressive symptoms revealed a significantly positive relation when interdependent self-construal was low (i.e., 1 SD below the mean), β = 0.51, *p* < 0.001, but not when it was high (i.e., 1 SD high the mean), *p* > 0.05 (see Figure 5).

Drawing from the above findings, bootstrapping was conducted to examine the mediation effects. Based on 5000 bootstrap samples with replacement, the 95% CI indicated that the standardized indirect effects of mindfulness on life satisfaction (CI: (−0.10, 0.05)) and depressive symptoms (CI: (−0.02, 0.02)) via the interaction term between interdependent self-construal and cognitive reappraisal included zeros. Likewise, the 95% CI indicated that the standardized indirect effects of mindfulness on life satisfaction (CI: (−0.04, 0.01)) and depressive symptoms (CI: (−0.05, 0.01)) via the interaction term between interdependent self-construal and expressive suppression included zeros. Therefore, although these emotion regulation strategies interacted with interdependent self-construal to predict the psychological outcomes, the moderated mediation effects were not supported. However, based on 5000 bootstrap samples with replacement, the 95% CI indicated that the standardized indirect effects of mindfulness via cognitive reappraisal on life satisfaction (CI: (0.04, 0.15)) and depressive symptoms (CI: (−0.13, −0.01)) did not include zeros, thereby supporting cognitive reappraisal alone as a mediator.

## 4. Discussion

The present study tested the moderating effect of interdependent self-construal between mindfulness, emotion regulation strategies, and psychological health. Supporting the mindful coping model [13], the findings revealed that regardless of the level of interdependent self-construal, cognitive reappraisal mediated between mindfulness and psychological health, including life satisfaction and depressive symptoms. Surprisingly, the findings also showed that expressive suppression was not related to mindfulness and, therefore, did not serve as a mediator. Moderation analyses further revealed differential strengths of association between mindfulness, emotion regulation strategies, and psychological outcomes as a function of interdependent self-construal.

The present findings echo with previous research [23,24] to show that cognitive reappraisal mediated between mindfulness and psychological adjustment, such as life satisfaction and depressive symptoms, regardless of the level of interdependent self-construal. According to the mindful coping model [13], mindful individuals are more likely to decenter themselves from stress appraisal and perceive their experiences with broadened attention and cognitive flexibility, which may enhance cognitive reappraisal [17]. Cognitive reappraisal, in turn, can promote positive psychological outcomes [10,11,20]. As such, mindful individuals are more likely to reappraise their negative experience, which fosters psychological adjustment.

Despite the mediation findings of cognitive reappraisal described above, post hoc simple slopes tests revealed differential strengths of association between variables as a function of interdependent self-construal. Notably, interdependent self-construal affected the relations between mindfulness, cognitive reappraisal, and life satisfaction. A significantly positive relationship was found between mindfulness and cognitive reappraisal for people with lower, but not higher, interdependent self-construal. Likewise, cognitive reappraisal enhanced life satisfaction more significantly for people with lower interdependent self-construal. For people with lower interdependent self-construal, individual factors such as mindfulness and cognitive reappraisal may have played important roles in people’s psychological functioning. For those with higher interdependent self-construal, perhaps psychological adjustment is less heavily affected by these factors. Other factors, such as concerns for other people, may have played a more important role instead (e.g., “I try to stay positive because I do not want other people to worry about me.”). As such, future research should further investigate psychological health as a function of both psychological and sociocultural factors.

Contrary to previous findings [17,18,19], the present study did not show a significant relationship between mindfulness and expressive suppression, regardless of the level of interdependent self-construal. Although mindful individuals are generally less judgmental, more accepting of their emotions, and less likely to suppress their outward expression [17], our data involving Chinese emerging adults indicated that mindfulness was not related to expressive suppression at all. Upon a closer examination of the literature, studies examining different facets of mindfulness did suggest that some, but not all, facets showed a significant relation to expressive suppression [36,37,38]. Specifically, in all three studies [36,37,38], “describing” was significantly associated with expressive suppression. However, “observing” was not related to expressive suppression across the studies, whereas “nonjudging” [36] and “nonreactivity” [37] was related to suppression in one of the three studies. Consequently, future research should further examine whether, why, and how different facets of mindfulness are linked to expressive suppression. Of note, although mindful individuals are prone to recognizing and accepting their emotions, which can motivate them to express their emotional experiences [17], whether they choose to express or suppress their emotions may be independent of this process. For example, people who accept their emotions non-judgmentally may suppress their expressions for other reasons, such as social desirability or out of concern for other people’s welfare. Consequently, future work is necessary to distinguish the relations between mindfulness, acceptance, and expression of emotions.

Despite the null findings between mindfulness and expressive suppression, the present findings did resonate with previous studies [25,26] in that the link between expressive suppression and depressive symptoms was stronger for people with lower interdependent self-construal. That is, expressive suppression was more strongly associated with depressive symptoms when Chinese emerging adults viewed themselves as less interdependent. Surprisingly, expressive suppression was negatively related to life satisfaction for those with higher, but not lower, interdependent self-construal. This finding is contrary to previous studies [26,28]. As a follow-up, future research should further examine how self-construal affects the relations between expressive suppression and various outcomes, such as subjective well-being, psychological distress, and physical and mental health.

The present findings should be interpreted in light of a few limitations. First of all, this study was cross-sectional. We cannot infer causality or direction of effects. Future studies should longitudinally examine these processes to establish a temporal sequence [39]. Studies should also use an experimental approach to establish causality. Second, our study exclusively relied on self-report questionnaires. Future studies may use various assessments, including structured interviews, physiological measures, and behavioral observations, to capture additional information. Third, previous research has examined different facets of mindfulness in relation to psychological functioning [2]. However, due to our small sample size, we only tested a path model with mindfulness as a manifest variable. Future studies should include a larger sample, such that different facets of mindfulness can be included as manifest indicators of mindfulness. In fact, based on power analysis [40], when alpha = 0.05, *df* = 3, desired power = 0.8, null RMSEA = 0.10, and alternative RMSEA = 0.04, the minimum *N* for this study should have been 650. Thus, future studies should include a much larger sample to replicate the present findings, with the addition of control variables such as age and sex. Fourth, the present emerging adult sample primarily of women limited the generalizability of the findings to other populations. Future studies may explore the relations in a diverse and gender-balanced sample via random sampling from the community. Fifth, we only examined the moderating effect of interdependent self-construal. Based on theories, researchers may examine the role of other moderators, such as face concern, which is a major concern in East Asian culture related to one’s social image and interpersonal relationship [41].

Taken together, the current study adds to the literature by examining the role of interdependent self-construal between mindfulness, emotion regulation, and psychological health. Findings highlighted the mediating role of cognitive reappraisal between mindfulness and psychological health. The findings are consistent with previous research [3,5,42], in that emotion regulation was closely connected to life satisfaction and depressive symptoms. Differential effects were also found, with interdependent self-construal as a moderator. Drawing from the present findings, future research should test theory-based models concerning mindfulness, emotion regulation, and psychological adjustment to verify their variability and applicability as a function of other factors, such as self-construal. Interventions aiming to promote psychological health among emerging adults merit future research investigations.

## Figures and Tables

**Figure 1 ijerph-18-00444-f001:**
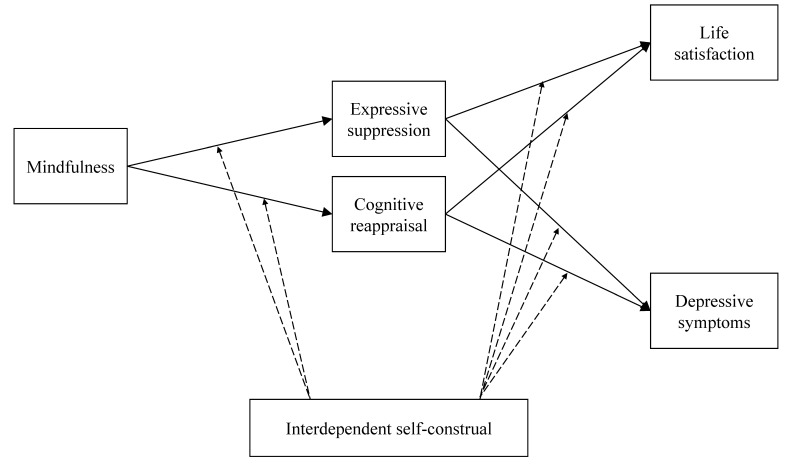
Conceptual model of mindfulness, emotion regulation, and psychological health, with interdependent self-construal as a moderator. In the analyses, the direct effects of mindfulness on life satisfaction and depressive symptoms were included to assess the mediation effects of emotion regulation strategies.

**Figure 2 ijerph-18-00444-f002:**
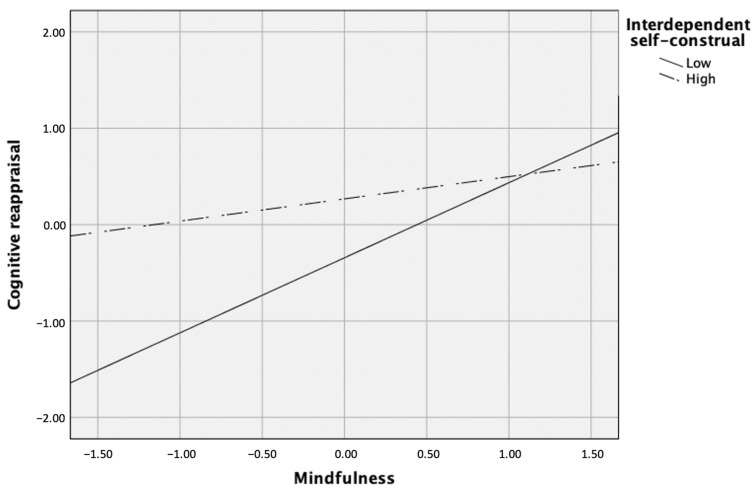
Interdependent self-construal as a moderator between mindfulness and cognitive reappraisal.

**Figure 3 ijerph-18-00444-f003:**
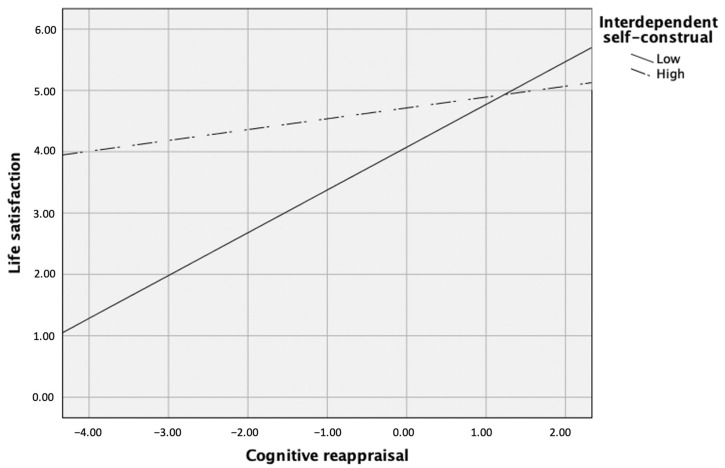
Interdependent self-construal as a moderator between cognitive reappraisal and life satisfaction.

**Figure 4 ijerph-18-00444-f004:**
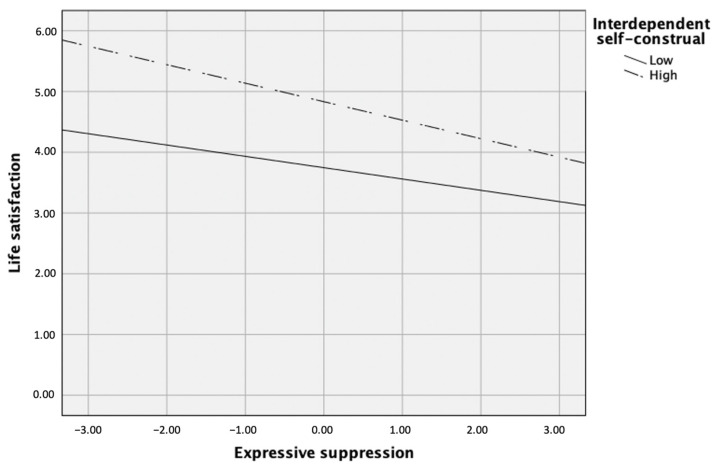
Interdependent self-construal as a moderator between expressive suppression and life satisfaction.

**Figure 5 ijerph-18-00444-f005:**
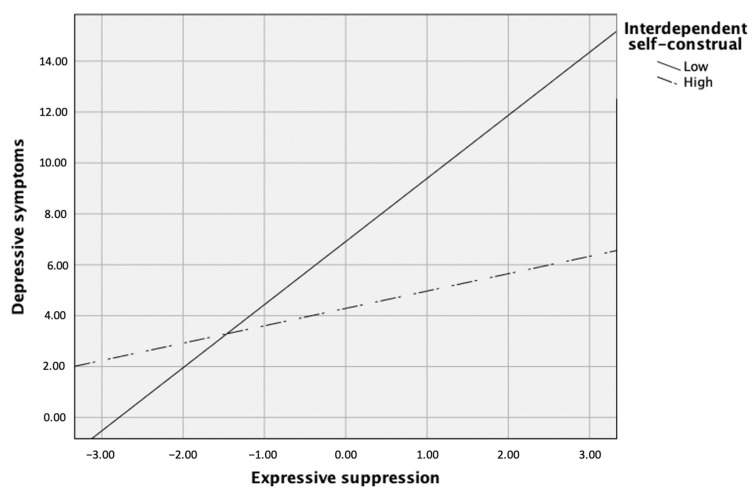
Interdependent self-construal as a moderator between expressive suppression and depressive symptoms.

**Table 1 ijerph-18-00444-t001:** Zero-order correlations, means, and standard deviations of the variables under study (*N* = 187).

Variable	(1)	(2)	(3)	(4)	(5)	(6)	(7)	(8)
(1) Sex	-							
(2) Age	0.00	-						
(3) Mindfulness	0.07	−0.10	-					
(4) Cognitive reappraisal	−0.01	−0.02	0.32 ***	-				
(5) Expressive suppression	−0.09	0.01	−0.08	−0.06	-			
(6) Life satisfaction	−0.02	0.08	0.30 ***	0.42 ***	−0.12	-		
(7) Depressive symptoms	0.11	−0.11	−0.47 ***	−0.35 ***	0.32 ***	−0.37 ***	-	
(8) Interdependent self-construal	0.10	−0.11	0.25 **	0.35 ***	0.05	0.38 ***	−0.25 **	-
M	-	21.16	2.85	5.27	3.58	4.31	5.51	3.82
SD	-	1.88	0.43	0.92	1.14	1.32	4.44	0.44

Note. ** *p* < 0.01, *** *p* < 0.001.

## Data Availability

The data presented in this study are available on request from the corresponding author. The data are not publicly available due to ethical restrictions.

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
