# Peer review of "Testing Interdependent Self-Construal as a Moderator between Mindfulness, Emotion Regulation, and Psychological Health among Emerging Adults"

_ijerph, 2021, doi:10.3390/ijerph18020444_

Round 1

Reviewer 1 Report

Review of ijerph-1037976

This article examines the moderating effect of interdependent self-construal on the mediation between trait mindfulness and well-being by emotional regulation strategy. It is found that cognitive reappraisal mediates the positive relationship between mindfulness and life-satisfaction and the negative relationships between mindfulness and depressive symptoms in individuals both high and low in interdependent self-construal. No significant mediation between mindfulness and well-being was found for expressive suppression, regardless of level of interdependent self-construal. Finally, the positive relationship between expressive suppression and depressive symptoms was weaker for individuals high in interdependent self-construal compared to individuals low in interdependent self-construal.

The paper is well-written and has a good theoretical framework. I am not convinced that a median-split of interdependent self-construal in an all Chinese sample is a valid measure of cultural differences. As such, I am not sure that what the authors did actually tests the theory of cultural differences moderating relationships among mindfulness, emotional regulation and well-being. Instead, the study design seems to better examine the moderating effect of interdependent self-construal levels within a single culture. As such, it could be argued that this “cultural value” differs significantly within all cultures and has predictive influences on emotions and well-being across cultures. 

Information on the power used to determine the sample size and the effect sizes and power of the results is also needed. The sample size seems small for the model tested. I also don’t understand why the decision to use a median split and test moderation of high/low interdependent self-construal groups was made. This variable significantly correlates with mindfulness, cognitive reappraisal, satisfaction with life, and depressive symptoms. Rather than create arbitrary groups, why not run a model that uses interdependent self-construal as a continuous moderator? 

Finally, this is a majority female sample, and there are gender differences in depressive symptoms and emotion regulation strategies. Even though sex was included in the model and found to not significantly correlate with any other variables, the amount of males included in the sample is so small that testing any effects of sex is tenuous. This issue needs to be addressed in the discussion of the results.

Minor points:

Page 2, line 53: Use “negatively” instead of “adversely”.

Page 2, line 57: Can you provide more information on the neural networks and/or brain structures that are found to be similarly active in mindfulness and cognitive reappraisal?

Page 2, line 67: Correct to “...participating in a schoolwide…”

Page 3-4, 2.2 Measures: The reporting of Cronbach’s alpha is phrased inconsistently throughout. Correct to “In this study, Cronbach’s alpha was equal to ##.##.”

Page 3, Line 137: The Cronbach’s alpha was lower than is considered an acceptable standard (.80) in social sciences. This should be addressed in the discussion.

Page 5, line 186: Correct to “...letting all paths differ by…”

Page 5, line 187: The 2 in “X^2” is not in superscript.

Page 6, Table 3: Please left-justify all the entries under the heading for column 1. There are no values presented in italics or bold as is stated in the Note.

Page 6, line 217: The “d” for the depressive symptoms CI is not in subscript.

Page 6, line 226, Correct to “... showing that the link between expressive suppression and depressive symptoms was …”

Page 7, lines 244-246: Other literature has looked at different facets of mindfulness. The self-report measure of mindfulness used in this study also has subscales. Why weren’t these examined?

Page 7, line 275: I don’t know what “face concern” is. Please provide a description.

Author Response

Comment:

The paper is well-written and has a good theoretical framework.

Response:

               Thank you for the positive comment.

Comment:

I am not convinced that a median-split of interdependent self-construal in an all Chinese sample is a valid measure of cultural differences. As such, I am not sure that what the authors did actually test the theory of cultural differences moderating relationships among mindfulness, emotional regulation and well-being. Instead, the study design seems to better examine the moderating effect of interdependent self-construal levels within a single culture. As such, it could be argued that this “cultural value” differs significantly within all cultures and has predictive influences on emotions and well-being across cultures. 

Response:

Thank you for the comment. We agree with Reviewer 1 that “cultural value” may not be the most appropriate term for this study. Where appropriate, we have replaced the term “cultural values” by “interdependent self-construal” to reflect the goals of this study.

Comment:

Information on the power used to determine the sample size and the effect sizes and power of the results is also needed. The sample size seems small for the model tested. I also don’t understand why the decision to use a median split and test moderation of high/low interdependent self-construal groups was made. This variable significantly correlates with mindfulness, cognitive reappraisal, satisfaction with life, and depressive symptoms. Rather than create arbitrary groups, why not run a model that uses interdependent self-construal as a continuous moderator? 

Response:

               Thank you for the comment. We agree with Reviewer 1 that using a median split of interdependent self-construal may not be the most appropriate. In addition, a much larger sample may be needed in conducting multigroup path analysis. Therefore, we have rerun the analyses with interdependent self-construal as a continuous moderator (see “Data Analysis” in lines 159-175).

Regarding the power analysis (Browne & Cudeck, 1993), when alpha = .05, df = 3, desired power = .8, null RMSEA = .10, and alternative RMSEA = .04, the minimum N should in fact be much larger, i.e., N = 650. Therefore, future studies should include a larger sample to replicate the present findings (see lines 332-335).

Reference:

Browne, M. W., & Cudeck, R. Alternative ways of assessing model fit. In K.A. Bollen & J.S. Long (Eds) Testing structural equation models 1993. Sage.

Comment:

Finally, this is a majority female sample, and there are gender differences in depressive symptoms and emotion regulation strategies. Even though sex was included in the model and found to not significantly correlate with any other variables, the amount of males included in the sample is so small that testing any effects of sex is tenuous. This issue needs to be addressed in the discussion of the results.

Response:

               Thank you for the comment. We have now addressed Reviewer 1’s comment as a limitation and a future direction (lines 334-338).

Comment:

Page 2, line 53: Use “negatively” instead of “adversely”.

Response:

Thank you for your comment. The word “adversely” has now been removed.

Comment:

Page 2, line 57: Can you provide more information on the neural networks and/or brain structures that are found to be similarly active in mindfulness and cognitive reappraisal?

Response:

Thank you for your comment, in the revised manuscript, we have added “mindfulness and cognitive reappraisal share some neural mechanisms, such as the regulating structures and the target region (Opialla et al,, 2015)” (lines 55-58).

Reference:

Opialla, S., Lutz, J., Scherpiet, S., Hittmeyer, A., Jäncke, L., Rufer, M., Holtforth, M. G. Herwig, U., & Brühl, A.B. Neural circuits of emotion regulation: A comparison of mindfulness-based and cognitive reappraisal strategies. Eur. Arch. Psychiatry Clin. Neurosci. 2015265(1), 45-55.

Comment:

Page 2, line 67: Correct to “...participating in a schoolwide…”

Response:

Thank you for your comment. The sentence has been revised (lines 66-67).

Comment:

Page 3-4, 2.2 Measures: The reporting of Cronbach’s alpha is phrased inconsistently throughout. Correct to “In this study, Cronbach’s alpha was equal to ##.##.”

Response:

               Thank you for the comment. We have standardized the phrase throughout the Method section.

Comment:

Page 3, Line 137: The Cronbach’s alpha was lower than is considered an acceptable standard (.80) in social sciences. This should be addressed in the discussion.

Response:

               Thank you for the comment. We have now added (lines 141-143), “[a]lthough the Cronbach’s alpha was less than 0.80 for expressive suppression, alpha larger than 0.7 are regarded as acceptable in psychological constructs (Kline, 2000).”

Kline, P. The handbook of psychological testing 2000. Psychology Press.

Comment:

Page 5, line 186: Correct to “...letting all paths differ by…”

Response:

Thank you for your comment. We have rewritten the Results section and phrase is gone.

Comment:

Page 5, line 187: The 2 in “X^2” is not in superscript.

Response:

Thank you for your comment. The “2” has been revised be superscript as “χ2” throughout the manuscript.

Comment:

Page 6, Table 3: Please left-justify all the entries under the heading for column 1. There are no values presented in italics or bold as is stated in the Note.

Response:

Thank you for your comment. All the entries under the heading for column 1 has been revised to left justified.

Comment:

Page 6, line 217: The “d” for the depressive symptoms CI is not in subscript.

Response:

Thank you for your comment. We have made the corrections accordingly.

Comment:

Page 6, line 226, Correct to “... showing that the link between expressive suppression and depressive symptoms was …”

Response:

Thank you for your comment. The sentence has been revised to “...the link between expressive suppression and depressive symptoms was...” (lines 313-314).

Comment:

Page 7, lines 244-246: Other literature has looked at different facets of mindfulness. The self-report measure of mindfulness used in this study also has subscales. Why weren’t these examined?

Response:

               Thank you for the comment. Due to our small sample size, we were unable to examine different facets of mindfulness as manifest indicators in the path model. We have now included this recommendation as a limitation and a future direction (lines 328-332).

Comment:

Page 7, line 275: I don’t know what “face concern” is. Please provide a description.

Response:

Thank you for your comment, the sentence has been revised to “such as face concern, which is a major concern in East Asian culture related to one’s social image and interpersonal relationship (Choi & Lee, 2002)” (lines 340-341).

Reference:

Choi, S. C., & Lee, S. J. Two-component model of chemyon-oriented behaviors in Korea: Constructive and defensive chemyon. J. Cross-Cult. Psychol. 200233(3), 332-345.

Reviewer 2 Report

The manuscript used 187 emerging adults to examine the moderating role of cultural values between mindfulness, emotion regulation, and psychological health in China. The results indicated that cognitive reappraisal mediated between mindfulness and psychological health and the link between expressive suppression and depressive symptoms was moderated by cultural values. The topic addressed in this manuscript is potentially important and may contribute to this line of research. However, I have concerns in the manuscript as listed below, that I hope can assist the manuscript.

  1. Literature: Some recent studies on positive effects of mindfulness on well-being can be added in the paper as below: Cheung, S., Xie, X., & Huang, C-C. (2020). Mind Over Matter?: Mindfulness, Income, Resilience, and Life Quality of Vocational High School Students in China. International Journal of Environmental Research and Public Health, 17, 5701.
  2. Sample: Sample was relatively small and not random selected from the population. Need to comment on the generalizability of the findings due to the sample limitation.
  3. Culture measure: This study used 14-item interdependent self-construal subscale of the Self-Construal Scale to measure culture may be debatable. If you do not use culture itself, why not use the term of self-construct rather than culture?
  4. Results: The authors wrote “the association between expressive suppression and depressive symptoms was significant for individuals with low interdependent self-construal (B = 1.78, SE = 0.35, p < 0.001)…” (line 190-192, page 5), but the number in Figure 1 was 1.82, not 1.78? Also, if I read the figure and text right, there was no moderate effect for cognitive reappraisal?
  5. Table 3: The table is hard to read, you have one estimate in the unstandardized estimate column and two estimates in standardized one. I guess the two numbers in the standardized column were for two different self-construal groups. However, there was no text to explain this, and the note of the table was not clear.
  6. Discussion: Culture, cultural values, and interdependent self-construal may be overlapped, but they are different concepts for me. I will recommend authors to use and measure the term consistently. I saw authors put a limitation regarding this concern, if so, why not label it as interdependent self-construal to avoid the confusion.
  7. Given moderator effect was emphasized on the paper tile, authors may need to elaborate on why interdependent self-construal did not have moderated effect between cognitive reappraisal and outcome variables.

Author Response

Comment:

Literature: Some recent studies on positive effects of mindfulness on well-being can be added in the paper as below: Cheung, S., Xie, X., & Huang, C-C. (2020). Mind Over Matter?: Mindfulness, Income, Resilience, and Life Quality of Vocational High School Students in China. International Journal of Environmental Research and Public Health, 17, 5701.

Response:

Thank you for your recommendation. The reference has now been added to the manuscript (line 39).

Comment:

Sample: Sample was relatively small and not random selected from the population. Need to comment on the generalizability of the findings due to the sample limitation.

Response:

Thank you for your comment. We have now added, “Future studies may explore the relations in a diverse and gender-balanced sample via random sampling from the community” (lines 336-338).

Comment:

Culture measure: This study used 14-item interdependent self-construal subscale of the Self-Construal Scale to measure culture may be debatable. If you do not use culture itself, why not use the term of self-construct rather than culture?

Response:

               Thank you for the comment. We have now used the term “self-construal” rather than “culture” throughout the manuscript.

Comment:

Results: The authors wrote “the association between expressive suppression and depressive symptoms was significant for individuals with low interdependent self-construal (B = 1.78, SE = 0.35, p < 0.001)…” (line 190-192, page 5), but the number in Figure 1 was 1.82, not 1.78? Also, if I read the figure and text right, there was no moderate effect for cognitive reappraisal?

Response:

               Thank you for the comment. As suggested by Reviewer 1, we have rerun the analyses. The Results are now presented in lines 180-262.

Comment:

Table 3: The table is hard to read, you have one estimate in the unstandardized estimate column and two estimates in standardized one. I guess the two numbers in the standardized column were for two different self-construal groups. However, there was no text to explain this, and the note of the table was not clear.

Response:

               Thank you for the comment. To improve clarity, we have removed Table 3 and presented the findings in the Results section.

Comment:

Discussion: Culture, cultural values, and interdependent self-construal may be overlapped, but they are different concepts for me. I will recommend authors to use and measure the term consistently. I saw authors put a limitation regarding this concern, if so, why not label it as interdependent self-construal to avoid the confusion.

Response:

Thank you for the comment. We agree with Reviewer 2 and replaced “culture” and “cultural values” by “interdependent self-construal” throughout the Discussion.

Comment:

Given moderator effect was emphasized on the paper title, authors may need to elaborate on why interdependent self-construal did not have moderated effect between cognitive reappraisal and outcome variables.

Response:

Thank you for the comment. We have further discussed the moderation findings throughout the Discussion section (lines 280-293; 312-321).

Round 2

Reviewer 1 Report

The authors have done a good job addressing my previous comments. I find the framing of the paper and the analysis of the data to be more appropriate in this revision. 

I have one minor comment remaining, the revision provided by the authors:

"Thank you for your comment, in the revised manuscript, we have added “mindfulness and cognitive reappraisal share some neural mechanisms, such as the regulating structures and the target region (Opialla et al,, 2015)” (lines 55-58)."

does not make sense to me. My background is in neuroscience, and I don't know what "the target region" refers to. I recommend providing a different description of what this are may be or removing it all together.

Reviewer 2 Report

The authors have addressed my previous concerns and the manuscript has improved substantially, good work, I do not have further comments and recommend acceptance of this manuscript.